# Rho-Associated Protein Kinase Activity Is Required for Tissue Homeostasis in the *Xenopus laevis* Ciliated Epithelium

**DOI:** 10.3390/jdb12020017

**Published:** 2024-06-11

**Authors:** Fayhaa Khan, Lenore Pitstick, Jessica Lara, Rosa Ventrella

**Affiliations:** 1Biomedical Sciences Program, College of Graduate Studies, Midwestern University, Downers Grove, IL 60515, USA; fayhaa.khan@midwestern.edu (F.K.); jessica.lara@midwestern.edu (J.L.); 2Department of Biochemistry and Molecular Genetics, College of Graduate Studies, Midwestern University, Downers Grove, IL 60515, USA; lpitst@midwestern.edu; 3Precision Medicine Program, College of Graduate Studies, Midwestern University, Downers Grove, IL 60515, USA

**Keywords:** lung cancer, ROCK, y-27632, *Xenopus laevis*, ciliated epithelium

## Abstract

Lung epithelial development relies on the proper balance of cell proliferation and differentiation to maintain homeostasis. When this balance is disturbed, it can lead to diseases like cancer, where cells undergo hyperproliferation and then can undergo migration and metastasis. Lung cancer is one of the deadliest cancers, and even though there are a variety of therapeutic approaches, there are cases where treatment remains elusive. The rho-associated protein kinase (ROCK) has been thought to be an ideal molecular target due to its role in activating oncogenic signaling pathways. However, in a variety of cases, inhibition of ROCK has been shown to have the opposite outcome. Here, we show that ROCK inhibition with y-27632 causes abnormal epithelial tissue development in *Xenopus laevis* embryonic skin, which is an ideal model for studying lung cancer development. We found that treatment with y-27632 caused an increase in proliferation and the formation of ciliated epithelial outgrowths along the tail edge. Our results suggest that, in certain cases, ROCK inhibition can disturb tissue homeostasis. We anticipate that these findings could provide insight into possible mechanisms to overcome instances when ROCK inhibition results in heightened proliferation. Also, these findings are significant because y-27632 is a common pharmacological inhibitor used to study ROCK signaling, so it is important to know that in certain in vivo developmental models and conditions, this treatment can enhance proliferation rather than lead to cell cycle suppression.

## 1. Introduction

Lung cancer is a heterogeneous disease that affects the airway epithelial cells, and it can be caused by genetic mutations as well as environmental impacts, most notably cigarette smoking. However, lung cancer can occur in those who have never smoked, indicating that there are a variety of additional risk factors associated with lung cancer pathogenesis [1,2]. This disease is one of the most frequently diagnosed cancers and is the leading cause of cancer worldwide [3,4]. Lung cancer is classified according to its histological subtype, and each of these subtypes often varies in lung location and cell of origin [5,6,7]. Lung adenocarcinoma (LUAD), a non-small-cell lung cancer (NSCLC), is the most common type of lung cancer and develops from the alveolar progenitor type 2 cells of the glandular epithelium in the lung periphery [8], whereas lung squamous-cell carcinoma (SqCC), another type of NSCLC, is thought to originate from the basal cells in the central airway and bronchi [5,9,10]. Small-cell lung cancer (SCLC) also originates from the central airways but is thought to be derived from neuroendocrine cells [11,12,13,14].

The basal cells within the central airway are considered to be the stem cell progenitors because they can differentiate into the secretory club cells, which can then give rise to both the mucus-secreting goblet cells and multiciliated cells (MCCs), as well as ionocytes, neuroendocrine cells, and tuft cells. During differentiation, basal cells undergo a complex and coordinated transcriptional process that directs them into these final cell fates. In addition to lung cancer, abnormal formation and functioning of these cells can lead to a variety of conditions, including COPD, asthma, cystic fibrosis, and primary ciliary dyskinesias [15,16,17].

One signaling pathway that has been shown to be central to cancer development is the RhoA/ROCK signaling pathway because its downstream signaling can control cell cycle progression, survival, cellular senescence, and migration [18]. RhoA is a GTPase protein that is activated by guanine exchange factors (GEFs) that exchange a GDP for a GTP. Active RhoA-GTP can then activate rho-associated protein kinase (ROCK), resulting in protumorigenic functions such as the cell cycle, survival, migration, and invasion [18,19,20]. Because of this, inhibition of ROCK has been considered as a molecular target to prevent tumor progression and metastasis [21]. One common inhibitor of ROCK is y-27632, which binds to the ATP binding site of ROCK, thus preventing downstream signaling [22]. However, in some conditions, ROCK inhibition with y-27632 has been shown to have contradictory results. For example, ROCK inhibition with y-27632 promoted proliferation and increased the clone-forming ability of airway epithelial basal cells. [23,24]. Similarly, the use of y-27632 has been shown to enhance the cloning efficiency of prostate stem cells and promote the proliferation of limbal epithelial cells and keratinocytes, as well as induce the invasiveness of colon cancer cells in a density-dependent manner [25,26,27,28]. These outcomes suggest that the precise role of RhoA/ROCK and y-27632 may be unclear and cell- and cancer-type specific. Therefore, further investigation of these opposing signaling consequences in different tissue cancer models is essential.

Here, we address the role of y-27632-induced ROCK inhibition using *Xenopus laevis* embryonic skin as a model system of ciliated epithelial development. Similar to the mammalian bronchiolar lung epithelium, *Xenopus laevis* embryonic skin contains secretory cells, MCCs, and ionocytes [29,30,31]. Also, the MCCs of *Xenopus* skin have been shown to be regulated by the same transcriptional cascades as mammalian lung, which, when altered, result in similar phenotypes [32]. For example, lateral inhibition by Notch and Delta ligands in p63-positive basal stem cells results in the expression of geminin coiled-coil domain-containing protein 1 (GEMC1) and multicilin (MCI) [33,34,35]. MCI leads to the activation of the motile ciliogenesis program through the activation of transcription factors like RFX2/3, C-myb, and FoxJ1, which promote the expression of core cilium and motile cilia genes [36,37,38,39]. Collectively, this transcriptional cascade results in the differentiation into MCCs that contain hundreds of motile cilia that beat in a metachronal synchronization on the epithelial surface [40]. Mutations and abnormal expression of these genes have been shown to result in motile ciliopathies like primary ciliary dyskinesia [41,42,43,44,45]. These similarities make *Xenopus* embryonic skin an ideal model system for a better understanding of ciliated epithelium development as well as mechanisms of SqCC pathogenesis and therapeutic responses. We found that treatment of *Xenopus* embryonic skin with y-27632 and other ROCK inhibitors caused abnormal tissue development due to epithelial hyperproliferation. Also, the abnormal tissue did not appear to have undergone an epithelial-to-mesenchymal transition (EMT). These findings offer insight into the diverse outcomes of ROCK inhibition and show that inhibition of the RhoA/ROCK pathway in this in vivo model system can promote proliferation and lead to epithelial tissue outgrowths.

## 2. Materials and Methods

### 2.1. Xenopus laevis Maintenance

*Xenopus laevis* were maintained according to the Midwestern University Institutional—Animal Care and Use Committee (IACUC) Protocol #3217. *Xenopus* were purchased from Xenopus 1 (Dexter, MI, USA). *Xenopus* eggs were obtained following ovulation stimulation, and in vitro fertilizations were performed to generate embryos, as previously described, or fertilized eggs were purchased from Xenopus 1 [46,47]. *Xenopus* embryos and tadpoles were maintained in 0.1× Marc’s Modified Ringer’s (MMR) solution and staged according to developmental data [48,49].

### 2.2. Drug Treatments

Drug treatments, including y-27632, y-33075, H-1152, and Fasudil (HA-1077), were used to treat *Xenopus laevis* embryos at varying concentrations as indicated in 0.1× MMR and refreshed every 2–3 days. For all experiments, embryos were treated beginning at stage 25/26 until the indicated stage; stage 42 occurred approximately two days after treatment; and stage 47 occurred approximately 4 days after treatment.

### 2.3. Fluorescent Imaging

*Xenopus* embryos/tadpoles were fixed at the indicated stage with 3% PFA/PBS, blocked in 10% goat serum/PBST (PBS + 0.1% Triton-X), followed by incubation in 5% goat serum/PBST with primary antibodies, as described in Table 1, and then Alexa Fluor secondary antibodies. Following antibody incubation, stains were used, as described in Table 2. When antibodies were not used, tadpole tails were stained after blocking. Then, tadpole tails were mounted between two coverslips using Fluoro-Gel (ThermoFisher Scientific, Waltham, MA, USA), as previously described [50].

For BrdU staining, 100 µM BrdU was added into the MMR with the treatments at stage 25/26 until the indicated stage. Following the fixation step, tadpoles were permeabilized in PBST for one hour at room temperature and then were incubated in 4 M HCl for 20 min at room temperature [51]. Then, the remaining staining procedure was followed as described above.

Following mounting, fluorescent whole-mount imaging was performed on the Nikon (Melville, NY, USA) A1R confocal microscope located in the Midwestern University Core Facility, Downers Grove, IL. Image processing and analysis were performed using FIJI software version v1.54f (ImageJ; National Institutes of Health, USA) [52].

### 2.4. Quantification of Abnormal Tissue Area

To quantify the abnormal tissue area, the freehand selection tool was used in FIJI to calculate the total area of the tail (mm^2^). Similarly, the area of all the outgrowths was determined and summed to obtain the total abnormal tissue area (mm^2^). The following calculation was then used to obtain the percent of the tail tissue occupied by outgrowths:$$Abrnomal\;Tissue\;Area\;\left(\%\right)=\left(\frac{Total\;Abnormal\;Tissue\;Area\;\left({mm}^{2}\right)}{Total\;Tail\;Area\;\left({mm}^{2}\right)}\right)\times 100\%$$

### 2.5. Statistical Analyses

For all experiments, embryos from three or more replicates were used. GraphPad Prism (Version 9.5; Boston, MA, USA) was used for data analysis. When two groups were compared, an unpaired *t*-test was used. When more than two groups with one treatment were compared, a one-way ANOVA with Tukey’s post hoc test was used. When there were more than two groups compared, a two-way ANOVA with Tukey’s post hoc test was used. The details for each statistical analysis performed are included in the corresponding figure legends.

## 3. Results

### 3.1. ROCK Inhibition Causes Abnormal Epithelial Growths

The RhoA/ROCK pathway has many downstream targets, including myosin light-chain phosphatase 1 (MYPT1), myosin regulatory light chain (MLC), and LIM kinases 1 and 2 (LIMK1 and LIMK2) that work together to alter cytoskeletal properties that can affect cellular motility, adhesion, and proliferation [18,19,20]. However, there is conflicting evidence about how inhibition of this pathway plays a role in maintaining and altering epithelial homeostasis [21,22,23,24,25,26,27,28]. To better understand this pathway, *Xenopus laevis* embryonic skin was used as a model of ciliated epithelium development and treated with the ROCK inhibitor, y-27632. This drug binds to the ATP-binding site of ROCK1 and ROCK2 in a competitive manner, resulting in the inhibition of the catalytic site and resultant downstream signaling [22,53,54]. This inhibition has been shown to result in a loss of stress fiber formation and tension, as well as an increase in cofilin-dependent actin depolymerization, ultimately disrupting the three-dimensional architecture of cells [55]. Specifically in *Xenopus*, treatment of embryos with y-27632 causes expansion of neural crest cells, but upon gross examination, *Xenopus* embryos and tadpoles treated with y-27632 can develop normally without any noticeable morphological differences [56,57].

To avoid any potential defects in neural crest and epithelial cell fate specification, *Xenopus* embryos were treated after neural development in the early tailbud stage, at approximately stage 25/26 [56]. This is also the time that the ciliated epithelium has differentiated to contain functional, beating MCCs [58]. The tadpoles were treated until stage 47, as this is when the epithelium undergoes drastic changes before metamorphosis [58,59]. During homeostasis, the epithelial cells along the *Xenopus* tail are arranged to form a discrete edge, as seen with actin staining (phalloidin) (Figure 1A). However, treatment with y-27632 (100 µM) disrupts this organization and causes epithelial outgrowths along the tail edge (Figure 1A). These growths include nucleus-containing cells and are often delineated from the main tissue by an enrichment of actin. To quantify the effect of y-27632 treatment, the percentage of tail area occupied by abnormal growths was calculated. This was achieved by taking the cumulative area of all the outgrowths for a given tail, shown as yellow dashed lines in the zoomed-in panels, and comparing that value to the total area of the tail. In normal tails, there are no outgrowths along the tail edge, resulting in an abnormal tissue area of zero percent (Figure 1B). However, y-27632 treatment caused a significant increase in abnormal tissue area. On average, one percent of the total tail area was occupied by tissue outgrowths (Figure 1B). These data highlight the importance of ROCK signaling to maintain tissue architecture and epithelial homeostasis.

To ensure that this effect is due to the inhibition of ROCK signaling rather than off-target effects, a variety of other ROCK inhibitors were tested, including y-33075, H-1152, and Fasudil (HA-1077) (Figure 2). All these inhibitors show different kinase inhibitor profiles, and Fasudil has a similar potency to y-27632, in contrast to y-33075 and H-1152, which have greater potencies [54,56,57,60,61,62,63]. Like y-27632, the other ROCK inhibitors also caused abnormal epithelial growths along the edge of the tail, as seen by the altered tissue morphology. These similar phenotypes further provide evidence that ROCK activity is essential for maintaining epithelial homeostasis.

### 3.2. ROCK Inhibition Impairs Tissue Homeostasis in a Concentration- and Time-Dependent Manner

It has previously been shown that 10 µM of y-27632 treatment can decrease ROCK kinase activity by over 95% [61]. To better delineate the concentration-dependence of ROCK inhibition on tissue phenotype, *Xenopus* embryos were treated with increasing concentrations of y-27632 beginning at stage 25/26 of development until stage 47. Treatment with 1 µM, 10 µM, and 100 µM of y-27632 induced the formation of abnormal tissue along the tail edge, as seen with phalloidin staining (Figure 3A). There was a significantly higher percentage of abnormal tissue in 10 µM and 100 µM y-27632 conditions relative to the 0 µM control (Figure 3B). However, as little as 1 µM was sufficient to alter epithelial homeostasis in some of the tadpole tails. This loss of homeostasis could be seen as early as stage 42 of development with 100 µM y-27632 treatment, which was approximately two days after initial treatment (Figure 4). These data suggest that even though ROCK inhibition is typically thought to maintain homeostasis, in this in vivo model of ciliated epithelium, ROCK inhibition with y-27632 impairs tissue homeostasis and instead results in the formation of tissue outgrowths along the tail edge.

### 3.3. ROCK Inhibition Promotes Tissue Hyperproliferation Resulting in Tissue Outgrowths

Due to the epithelial outgrowths, it led us to propose that inhibition of ROCK was leading to uncontrolled cellular proliferation. To test this, BrdU was incorporated along with y-27632 to detect proliferating cells during development. This would allow for all the proliferating cells from stage 25/26 to be labeled with the BrdU thymidine analog [64]. During normal development, proliferating cells were mainly localized throughout the *Xenopus* notochord and the epithelial tail edge (Figure 5). However, ROCK inhibition with 100 µM y-27632 caused an abundant increase in cellular proliferation, as seen with the increased intensity and distribution of BrdU staining. BrdU had been evenly incorporated into most of the cells throughout the epithelium, as well as through the notochord (Figure 5). It is likely that this y-27632-induced hyperproliferation leads to the abnormal tissue outgrowths along the *Xenopus* tail edges prior to undergoing terminal differentiation.

### 3.4. ROCK-Inhibitor-Induced Growths Have Not Undergone EMT

One potential identity that these outgrowths may have taken on, in addition to proliferation, is a migratory phenotype. For these epithelial cells to undergo migration, they would need to initiate an epithelial-to-mesenchymal transition (EMT). This process is characterized by a loss of epithelial markers, such as E-Cadherin, and an increase in mesenchymal markers, like N-Caherin [65,66,67,68]. In the normal *Xenopus* epithelium, E-Cadherin is expressed at the cellular junctions, whereas N-Cadherin is diffusely expressed throughout the tissue (Figure 6). Similarly, with 100 µM y-27632 treatment, this expression pattern is maintained in the epithelial outgrowths, where there is still E-Cadherin expression at the cell borders, although not as strongly as the surrounding tissue. Additionally, the epithelial outgrowths still fail to express N-Cadherin (Figure 6). This leads us to conclude that these epithelial outgrowths induced from y-27632 treatment have not transitioned into a migratory state but rather remain in a hyperproliferative state.

## 4. Discussion

Inhibition of RhoA/ROCK signaling has previously been shown to have both pro- and antitumorigenic outcomes [23,24,25,26,27,28]. Here, we show that the inhibition of ROCK with y-27632 caused epithelial outgrowths in *Xenopus* embryonic skin as early as 48 h post-treatment along the edges of *Xenopus* embryonic skin, which can serve as a model for ciliated epithelium. These findings may suggest a unique signaling pathway in ciliated epithelium, where RhoA/ROCK is an antitumorigenic signaling pathway. So, when ROCK is inhibited with y-27632, it results in epithelial tissue outgrowths.

The y-27632-induced tissue becomes hyperproliferative, leading us to question the identity of this altered tissue. Tumor plasticity is a major driver of tumorigenesis, and the dedifferentiation of mature cells increases proliferative cell properties [69,70,71]. Our BrdU results lead us to hypothesize that the differentiated cells of the ciliated epithelium can undergo dedifferentiation, resulting in enhanced proliferation. This increase in proliferation may be through a PTEN/PIP3/Akt signaling network. Activating mutations in Ras have been shown to decrease ROCK signaling due to the increased interaction between RhoA and its GDP dissociation inhibitor, RhoGDI. This has been shown to decrease the dephosphorylation of PIP3 by PTEN, causing sustained activation of Akt and subsequent proliferation [72]. Because of the increased proliferation, the epithelium is no longer in a state of homeostasis, and the cells along the tail edge get pushed out of the perimeter due to the interior proliferating cells. Because of this, the cells along the tail edge form these tumor-like structures. However, it is also possible that instead of dedifferentiation, it is the progenitor cells in the epithelium that are responsible for this hyperproliferation. For example, ROCK may play a role in the inhibition of Wnt-mediated basal cell proliferation. So, inhibition of ROCK may lead to chronic Wnt signaling and subsequent basal cell hyperplasia [35]. Another potential progenitor capable of contributing to this hyperproliferation is the bipotent neuromesodermal progenitor cells (NMPs). These cells are in the tailbud and can generate caudal spinal cord neuroectoderm and paraxial mesoderm tissues [73]. Interestingly, the proliferation of NMPs is also dependent on Wnt signaling, leading to the possibility that if ROCK inhibition increases Wnt, it may lead to the increased proliferation of basal cells and NMPs [74,75]. This proposed mechanism may explain the wide distribution of BrdU staining in the y-27632-treated tails.

It is also possible that the abnormal tissue outgrowths are early-stage cysts. Lung cysts are known to be formed in a variety of cystic lung diseases, including lymphangioleiomyomatosis, Langerhans cell histiocytosis, lymphoid interstitial pneumonia, Birt-Hogg–Dubé syndrome, amyloidosis, as well as metastatic cancer in the distal airways [76,77]. Previous studies in lung epithelium development have emphasized that when there are defects in essential cellular processes, like Rho/ROCK signaling, it could lead to abnormal lung development and cyst formation [78,79]. Blocking downstream ROCK signaling with y-27632 may affect underlying pathways that influence further cell development, like cell polarity, mechanical force production, and apoptosis, which collectively could lead to cyst formation. Most notably, inhibition of ROCK with y-27632 has been shown to inhibit MLC phosphorylation and tension generation, resulting in abnormal lung morphogenesis and lack of epithelial bud formation [78]. It has been shown that a regulator upstream of this pathway is Yap. Loss of Yap decreases the expression of the RhoGEF, Arhgef17, leading to decreased activation of the RhoA/ROCK pathway and, ultimately, a loss of pMLC-induced tension. Reduced activation of the Yap/Rho/ROCK/pMLC pathway results in omnidirectional lung outgrowth rather than the directional outgrowth that is required to form lung buds [79]. The loss of cellular organization that is seen in the Yap-deficient lung cysts is reminiscent of the abnormal tissue outgrowths that result from ROCK inhibition, further suggesting the importance of ROCK signaling in maintaining homeostasis due to its central role in mechanotransduction pathways. A more in-depth investigation into this mechanotransduction pathway would give insight into this possibility. If this were the case, this model system may prove useful in studying cystic lung diseases.

The opposing outcomes of ROCK signaling as both tumor-suppressive and pro-oncogenic highlight a need for a better understanding of what drives these differences [18,19,20,21,23,24,25,26,27,28]. ROCK signaling has been shown to be upregulated in KRAS mutant NSCLC cell lines, animal models, and tumor tissues derived from patients, and it is thought that the increased stiffness of the extracellular matrix of the tumor, combined with hypoxia, activates the RhoA/ROCK pathway [21]. This leads to the possibility that differences in tension and mechanotransduction are the discriminating factors in determining the outcomes of ROCK signaling. This would suggest that in this in vivo model system, lack of tumor-induced stiffness makes ROCK a tumor suppressor, and then increased stiffness converts ROCK into an oncogene. Another intriguing discriminator is oxygen levels. It has been shown that decreased oxygen levels can increase RhoA activity due to the upregulation of galectin-3, resulting in increased migration. Interestingly, the knockdown of galectin-3 was able to rescue hypoxia-induced upregulation of ROCK activity and subsequent tumor cell motility [80]. The lack of a hypoxic environment may make ROCK signaling essential for homeostasis in this in vivo model system. This potential hypoxia/galectin-3/Rho/ROCK pathway may be used as a predictive biomarker for when ROCK inhibition may be effective in cancer treatment. For example, ROCK inhibition may be an effective therapeutic strategy in tumors that have an upregulation of galectin-3 due to hypoxia, whereas normoxic tumors may not benefit from this treatment strategy and may potentially worsen tumor-like phenotypes, similar to what we have seen in this in vivo model system. Additional studies examining how gradients in stiffness and oxygen levels could alter ROCK signaling may offer a predictive strategy to determine when ROCK inhibitor therapies would be an effective therapeutic strategy in cancer treatment.

## 5. Conclusions

Even though there are a variety of therapeutic approaches for lung cancer, there are still many lung cancer subtypes where treatment remains elusive. Here, we show the characterization of a ROCK-inhibitor-induced model of abnormal ciliated epithelial development in *Xenopus laevis* embryonic skin. In this model, we found that treatment with y-27632 induced tissue outgrowths along the tail edge. Normally, ROCK inhibitors are thought to be a beneficial treatment to inhibit proliferation [21]. However, our in vivo studies using the ciliated epithelium of *Xenopus laevis* tadpole showed that ROCK inhibition can lead to an opposing cellular outcome characterized by increased proliferation and the formation of epithelial outgrowths. This study highlights the importance of better understanding the Rho/ROCK signaling pathway in different tissue types and conditions.

## Figures and Tables

**Figure 1 jdb-12-00017-f001:**
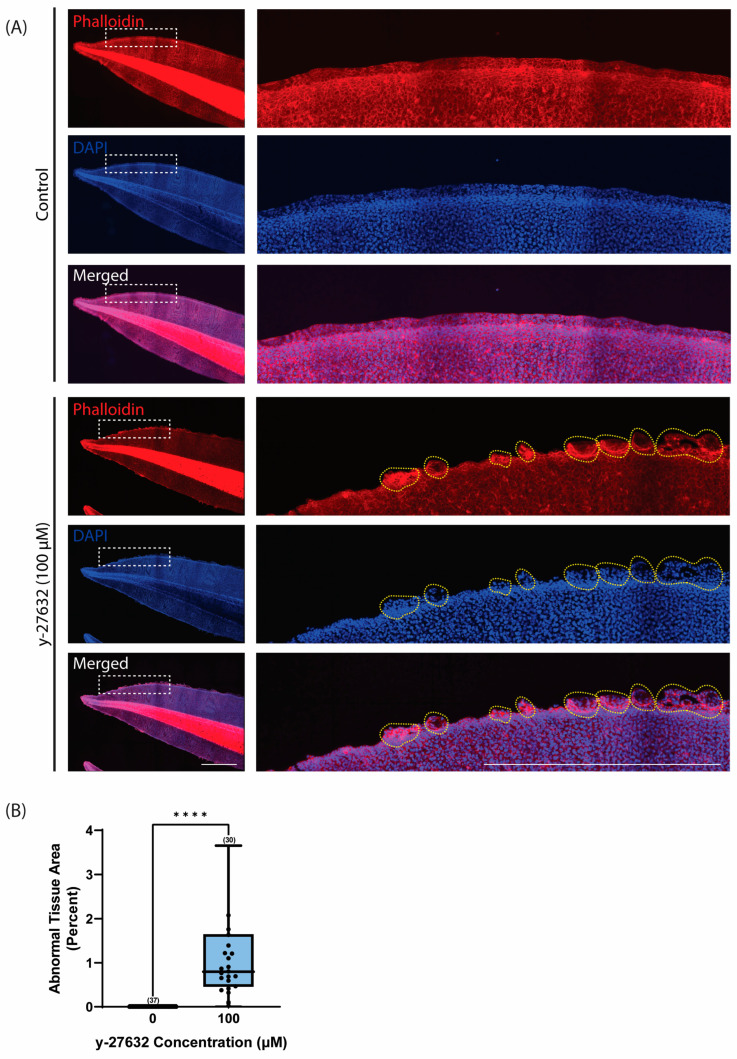
ROCK inhibition with y-27632 causes abnormal epithelial outgrowths. (**A**) Representative images of phalloidin (red) and DAPI (blue) staining in *Xenopus* tails at stage 47 of development after treatment with 0 µM and 100 µM of y-27632. Images on the right are magnified regions of the boxed area on the left. The yellow dashed lines in the 100 µM y-27632 condition show tissue outgrowths that were quantified for the cumulative outgrowth area relative to the total tail area. Scale bars represent 1 mm. (**B**) Quantification of abnormal tissue area (percent) at stage 47 of development after treatment with 0 µM and 100 µM of y-27632 (*n* ≥ 6 embryos per treatment). Box-and-whisker plots show mean with minimum and maximum (whiskers) and 25th–75th percentiles (boxes). An unpaired *t*-test was used for statistical analysis (**** *p* ≤ 0.0001). The number of tadpoles analyzed for each condition is given in parentheses, and each tail is represented as an individual data point.

**Figure 2 jdb-12-00017-f002:**
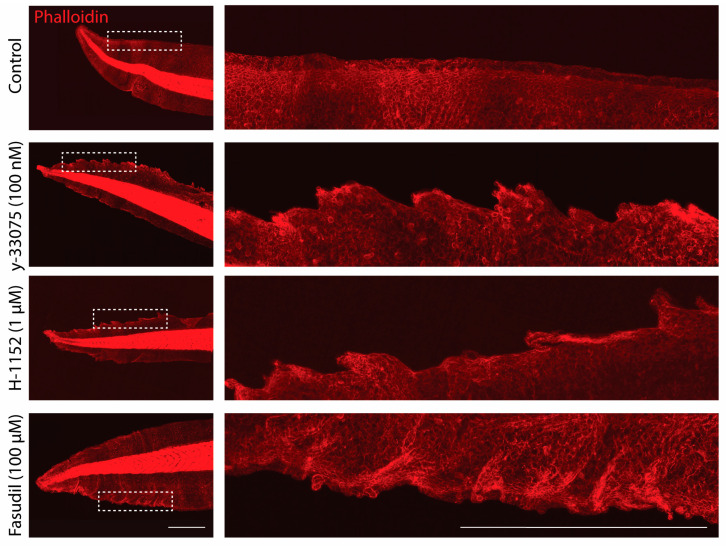
ROCK inhibition disrupts epithelial homeostasis. Representative images of phalloidin (red) staining in *Xenopus* tails at stage 47 of development after treatment with control, 100 nM y-33075, 1 µM of H-1152, and 100 µM Fasudil. Images on the right are magnified regions of the boxed area on the left. Scale bars represent 1 mm.

**Figure 3 jdb-12-00017-f003:**
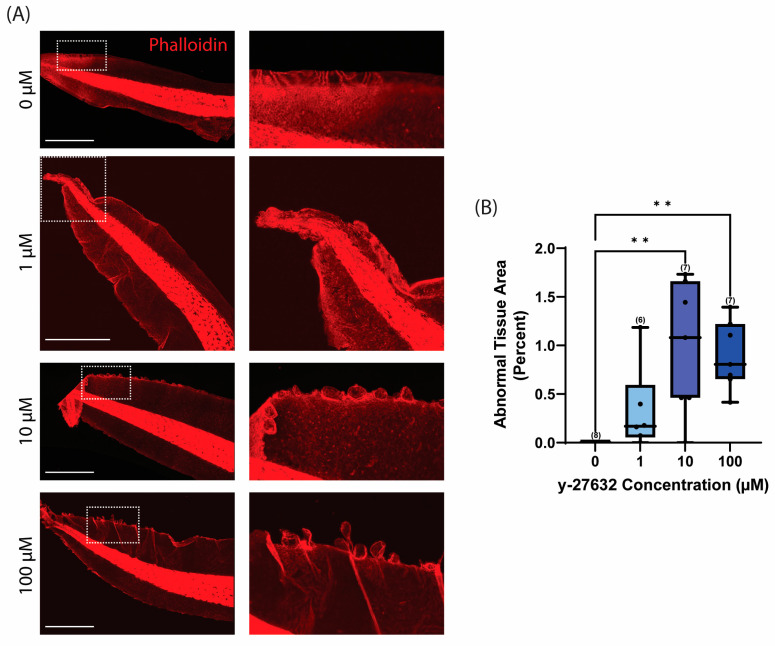
y-27632 induces abnormal tissue formation in a dose-dependent manner. (**A**) Representative images of phalloidin staining (red) in *Xenopus* tails at stage 47 of development after treatment with the given concentration of y-27632 beginning at stage 25/26 of development. Images on the right are magnified regions of the tissue in the boxed area on the left. Scale bars represent 1 mm. (**B**) Quantification of abnormal tissue area at stage 47 in *Xenopus* tails at given y-27632 concentrations. Box-and-whisker plots show mean with minimum and maximum. A one-way ANOVA with Tukey’s post hoc test was used for statistical analysis (** *p* ≤ 0.01). The number of tadpoles analyzed for each condition is given in parentheses, and each tail is represented as an individual data point.

**Figure 4 jdb-12-00017-f004:**
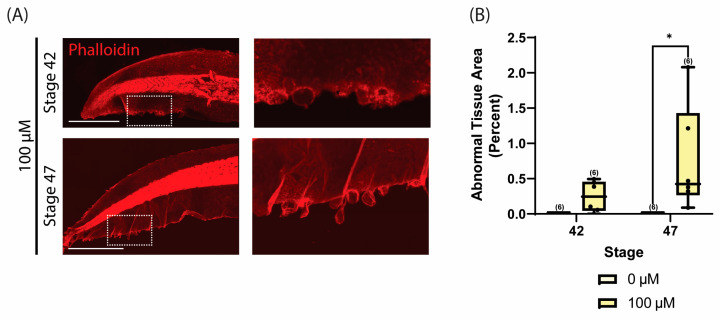
y-27632 induces abnormal tissue formation in a time-dependent manner. (**A**) Representative images of phalloidin staining (red) in *Xenopus* tails at stage 42 and stage 47 of development after treatment with 100 µM of y-27632 beginning at stage 25/26 of development. Images on the right are magnified regions of the tissue outgrowths in the boxed area on the left. Scale bars represent 1 mm. (**B**) Quantification of abnormal tissue area at stages 42 and 47 in *Xenopus* tails at given y-27632 concentrations. Box-and-whisker plots show mean with minimum and maximum. A two-way ANOVA with Tukey’s post hoc test was used for statistical analysis (* *p* ≤ 0.05). The number of tadpoles analyzed for each condition is given in parentheses, and each tail is represented as an individual data point.

**Figure 5 jdb-12-00017-f005:**
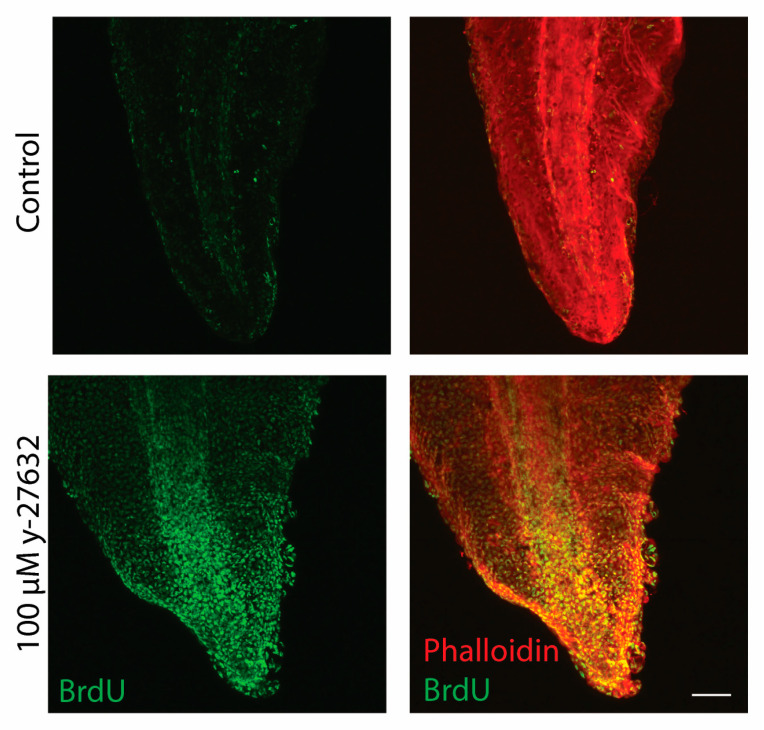
y-27632 increases BrdU incorporation. Representative images of BrdU (green) and phalloidin (red) staining in *Xenopus* tails at stage 47 of development after being treated with 100 µM of y-27632 and BrdU beginning at stage 25/26 of development. The scale bar represents 100 µm.

**Figure 6 jdb-12-00017-f006:**
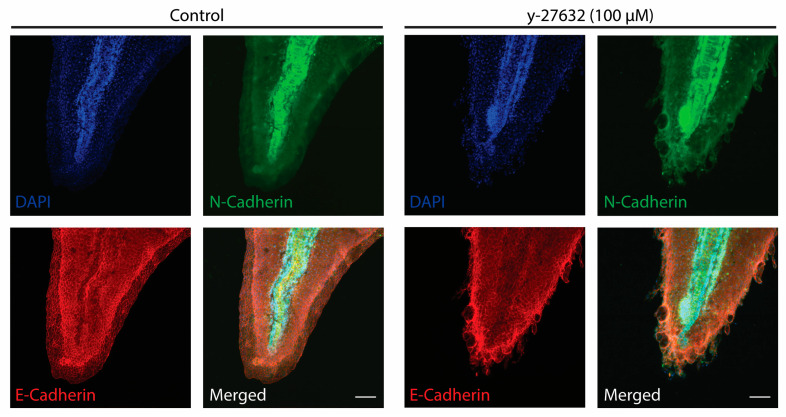
y-27632-induced epithelial outgrowths lack markers of EMT. Representative images of DAPI (blue), N-Cadherin (green), and E-Cadherin (red) staining in *Xenopus* tails at stage 47 of development after being treated with 100 µM of y-27632 beginning at stage 25/26 of development. Scale bars represent 100 µm.

**Table 1 jdb-12-00017-t001:** Information on antibodies used for immunofluorescent staining.

Antibody (Species)	Concentration	Company; Catalog Number
BrdU (mouse)	1:1000	Proteintech (Rosemont, IL, USA); 66241-1-AP
E-Cadherin (mouse)	1:200	DSHB* (Iowa City, IA, USA); D3
N-Cadherin (Rat)	1:50	DSHB*MNCS2

* DSHB: Developmental Studies Hybridoma Bank.

**Table 2 jdb-12-00017-t002:** Information on stains used for fluorescent imaging.

Stain (Cellular Structure Labeled)	Concentration	Company; Catalog Number
Phalloidin (Actin)	1:300	Cytoskeleton (Denver, CO, USA); PHDH1, PHDN1
PNA (Mucus)	1:300	ThermoFisher Scientific; L32460
DAPI (Nucleus)	1:2000	ThermoFisher Scientific; EN62248

## Data Availability

Data are available from the authors of this study upon request.

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
