# Peer review of "Rho-Associated Protein Kinase Activity Is Required for Tissue Homeostasis in the Xenopus laevis Ciliated Epithelium"

_jdb, 2024, doi:10.3390/jdb12020017_

Round 1
Reviewer 1 Report
Comments and Suggestions for Authors

Reviewer 2 Report
Comments and Suggestions for Authors
In general, the article is simple, the set of approaches and methods is very modest, but the article is written simply, and the conclusions are clear.
However, there are a number of comments.
Lanes 147-151 "the percentage of tail area occupied by abnormal growths was calculated"
Unclear wording, percentage of what in relation to what? What do you mean that abnormal growths appear only on some small part of the tail, and the ratio of the area with abnormalities to the total area of the tail is measured? Not clear. Needs reformulation.
And it would be nice to somehow reflect this in the corresponding picture (fig.1).
Similarly with Lanes 215-217, the meaning is not obvious from the text.
Lane 224 Caption for Fig. 5: written A instead of B?
Lane 257 reamining => remining? Typo?
Lane 291-303 we need to place the discussion of mechanotransduction in a more general context: what molecular mechanisms might be involved? The same applies to the part about the role of oxygenation.
In addition, the possible molecular mechanisms of the influence of ROCK on the observed downstream processes should be mentioned in the “discussion” section.
It is known that ROCK1 and ROCK2 are expressed at the considered stages in various tissues and organs (brain, eyes, cement gland, heart primordium, placodes), but the authors do not even mention the developmental anomalies of these organs when treated with y-27632.
Round 2
Reviewer 1 Report
Comments and Suggestions for Authors
The manuscript of Khan F et al, has been improved.Required Controls and informations about MCE have been added. I agree with the authors concerning the deletion of (prevuous) Fig. 5 (Figure 5: y-27632-induced epithelial outgrowths lack markers of differentiation), as this data deserves more attention, and deep and rigorous quantitative analyses.